# Brillouin Scattering Study of Ferroelectric Instability of Calcium–Strontium–Barium Niobate Single Crystals

**DOI:** 10.3390/ma16062502

**Published:** 2023-03-21

**Authors:** Seiji Kojima, Md Aftabuzzaman, Jan Dec, Wolfgang Kleemann

**Affiliations:** 1Division of Materials Science, University of Tsukuba, Tsukuba 305-8573, Japan; 2Department of Physics, Pabna University of Science and Technology, Pabna 6600, Bangladesh; 3Institute of Materials Science, University of Silesia, PL-40-007 Katowice, Poland; 4Angewandte Physik, University of Duisburg-Essen, D-47048 Duisburg, Germany

**Keywords:** Brillouin scattering, ferroelectric, relaxor, acoustic modes, elastic properties, slowing down

## Abstract

Uniaxial ferroelectrics with tetragonal tungsten bronze structure are important functional materials with photorefractive, electrooptic, piezoelectric, and pyroelectric properties. Sr*_x_*Ba_1−*x*_Nb_2_O_6_ (SBN100*x*) with *x* > 50 is known as a typical uniaxial relaxor ferroelectric, while Ca*_x_*Ba_1−*x*_Nb_2_O_6_ (CBN100*x*) undergoes nearly normal ferroelectric phase transitions. Single crystals of CSBN100*x* = [*x*(CBN28) + (1 − *x*) (SBN61)] = *x*Ca_0.28_Ba_0.72_Nb_2_O_6_ + (1 − *x*) Sr_0.61_Ba_0.39_Nb_2_O_6_ with nominal *x* = 0.00, 0.25, 0.50, 0.75, and 1.00 were studied to clarify the dynamical properties at the crossover from relaxor (*x* = 0) to normal (*x* = 1) ferroelectric behavior. The longitudinal acoustic (LA) and transverse acoustic (TA) modes and a central peak (CP) related to the relaxation process of polarization fluctuations along the polar *c*-axis were studied in uniaxial ferroelectric CSBN single crystals as a function of temperature via Brillouin scattering spectroscopy. A CBN28 (*x* = 1.00) crystal shows the sharp elastic anomaly of the LA mode in the gigahertz range toward Curie temperature, *T*_c_. However, those of CSBN25 (*x* = 0.25) and SBN61 (*x* = 0.00) crystals show diffusive anomalies due to stronger random fields. The relaxation time determined from the width of a CP shows a critical slowing down in the vicinity of *T*_c_. The elastic anomaly and slowing down of relaxation time of CSBN100*x* crystals become diffusive in the vicinity of *T*_c_ as the CBN28 content decreases. The origin of the crossover from relaxor to normal ferroelectric phase transitions is discussed in terms of the difference in the A1 and A2 sites’ occupancies.

## 1. Introduction

Ferroelectricity is defined by the existence of a spontaneous polarization, the direction of which is switchable by an external electric field [1]. Uniaxial ferroelectric materials with a tetragonal tungsten bronze (TTB) structure are technologically important for optical applications involving electro-optic, nonlinear optic, photorefractive, pyroelectric, and piezoelectric properties [2]. In light of recent environmental problems, Pb-free ferroelectrics and their functional properties have become important [3]. In TTB ferroelectrics, the direction of spontaneous polarization is restricted to the polar *c*-axis, which is why they are also called uniaxial ferroelectrics [4]. The structural formula of TTB ferroelectrics is expressed by (A1)_2_(A2)_4_(C)_4_-(B1)_2_(B2)_8_O_30_, with corner-sharing distorted BO_6_ octahedra as shown in Figure 1. The smallest C site is occupied only by Li, such as in K_3_Li_2_Nb_5_O_15_(KLN). In ferroelectric Ba_2_NaNb_5_O_15_ (BNN), which is well known by its excellent second harmonic generation, all the A1 and A2 sites are occupied by Ba^2+^ and Na^1+^ ions, respectively. BNN belongs to the filled TTB ferroelectrics due to their complete occupation of A1 and A2 sites and undergoes a normal ferroelectric phase transition with a sharp dielectric anomaly in the vicinity of a ferroelectric Curie temperature, *T*_c_, which is the highest among TTB ferroelectrics [2,5]. In contrast, the A1 sites of Sr*_x_*Ba_1−*x*_Nb_2_O_6_ (SBN100*x*) are occupied in part by Sr^2+^ ions, and the A2 sites are occupied in part by both Ba^2+^ and Sr^2+^ ions. Since 1/6 (A1 + A2) sites remain unoccupied, it belongs to the so-called unfilled (open) TTB ferroelectrics due to the incomplete occupancy of A1 and A2 sites. The empty A1 and/or A2 sites cause the charge disorder and are the main sources of quenched random fields (RFs), which enhance the relaxor features characterized by the frequency dispersion of dielectric susceptibility and diffusive phase transition [6,7]. In SBN, the strength of RFs increases as the Sr content of smaller (1.12 Å) Sr^2+^ ions increases, and SBN has attracted much attention as a typical uniaxial relaxor ferroelectric [8].

The disadvantage of SBN in application is the relatively low Curie temperatures. The Ca*_x_*Ba_1−*x*_Nb_2_O_6_ (CBN100*x*) compounds also belong to the unfilled TTB structure and show quite similar physical properties to SBN, while their Curie temperatures are much higher than those of SBN [9]. Therefore, the excellent optical and ferroelectric properties of CBN make them potential candidates for applications at relatively high temperatures [10]. Recently, CBN nanopowders were synthesized. Their analysis of the various optical properties, especially the photorefractive effect, suggests that CBN nanopowders can be potentially applied for ultrahigh-density optical data storage [11]. In CBN, most of the smaller (0.99 Å) Ca^2+^ ions occupy A1 sites, whereas the relatively larger (1.34 Å) Ba^2+^ ions predominantly occupy the A2 sites [12,13]. Thus, in contrast to SBN with strong RFs, the lower degree of disorder of the Ca^2+^ and Ba^2+^ ions of CBN causes the weak RFs, and the diffusive nature of CBN is weaker than that of SBN [14]. In the Brillouin scattering study of CBN28, the intense central peak (CP) caused by polarization fluctuations along the *c*-axis was clearly observed in the vicinity of *T*_c_. The relaxation time determined by the CP width clearly shows critical slowing down towards *T*_c_, reflecting a weakly first-order phase transition under weak RFs [15].

Within the quasi-ternary CaNb_2_O_6_-SrNb_2_O_6_-BaNb_2_O_6_ system, selected Ca_x_Sr_y_Ba_1−x−y_Nb_2_O_6_ (CSBN) compounds were grown via the Czochralski method in a tungsten–bronze-type structure [16]. The solid solution of CSBN is technologically important due to the coexistence of the high *T*_c_ of CBN and excellent functionality of SBN. In fundamental science, the crossover of CSBN from weak RFs of CBN with a nearly normal ferroelectric nature to strong RFs of SBN with a relaxor nature is very interesting regarding control of the strength of RFs only by Ca content. Since there are three kinds of cations, Ca^2+^, Sr^2+^, and Ba^2+^, with increasing ionic radii occupying A1 and A2 sites, the degree of freedom to control physical properties increases. The Curie temperatures of *x*CBN28-(1 − *x*) SBN61 (CSBN100*x*) are shown in Figure 2. As the Ca content increases in CSBN, *T*_c_ monotonically increases, and the ferroelectric phase transition becomes less diffusive due to the decrease of disorder, which indicates the suppression of the RFs [17,18]. In addition, it is worth mentioning here that the above composition formula of the CSBN system may be expressed in a more compact fashion: (i) *x* = 0, Sr_0.61_Ba_0.39_Nb_2_O_6_ for SBN61, (ii) *x* = 0.25, Ca_0.07_Sr_0.458_Ba_0.472_Nb_2_O_6_ for CSBN25, (iii) *x* = 0.50, Ca_0.14_Sr_0.305_Ba_0.555_Nb_2_O_6_ for CSBN50, (iv) *x* = 0.75, Ca_0.21_Sr_0.153_Ba_0.637_Nb_2_O_6_ for CSBN75, and (v) *x* = 1.00, Ca_0.28_Ba_0.72_Nb_2_O_6_ for CBN28.

Brillouin scattering is the inelastic light scattering by thermally excited sound waves and has been used as a non-contact and non-destructive method to obtain elastic constants in the gigahertz range [19]. In the present study, the elastic properties and dynamical instability of CSBN100*x* crystals were investigated by using Brillouin scattering spectroscopy.

## 2. Experimental Methods

CSBN100*x* single crystals were grown via the Czochralski method [16] for the nominal compositions, *x* = 0.00, 0.25, 0.5, 0.75, 1.00. The CSBN single crystals were grown at the Institute of Electronic Materials Technology (Poland) under the guidance of Prof. T. Lukasiewicz. Their structural measurements were published [20,21]. The real composition of the crystals obtained was checked with the use of ICP-OES (inductively coupled plasma–optical emission spectroscopy) method and the result was published [22]. It has appeared that the real composition was, within uncertainties, satisfactorily close to the established one. Single crystalline plates were cut along [100] (*a*-plate) and [001] (*c*-plate) with optically polished 5 × 5 mm^2^ surfaces and 1 mm thickness. Brillouin scattering spectra were measured at the back scattering geometry using a high-contrast 3 + 3 passes tandem Fabry–Perot interferometer, as shown in Figure 3 [19]. The exciting source was a diode-pumped solid state (DPSS) laser with a wavelength of 532 nm and a power of 100 mW. Scattered light was detected using a photon counting system. The specimen’s temperature was controlled using a cooling/heating stage (Linkham, THMS600, Salfords, UK) with a stability of ±0.1 °C. Using an *a*-plate, longitudinal acoustic (LA) and transverse acoustic (TA) modes–which propagate along the *a*-axis—and the CP of polarization fluctuations along the ferroelectric *c*-axis were measured. Using a *c*-plate, LA and TA modes—which propagate along the *c*-axis—were measured.

## 3. Results and Discussion

### 3.1. Elastic Anomaly of LA Modes

The temperature dependences of Brillouin scattering spectra of a CSBN50 crystal measured at the backward scattering geometry using *a*- and *c*-plates are shown in Figure 4 and Figure 5, respectively. The direction of the wave vector of a scattered phonon is parallel to the *a*-axis for an *a*-plate and to the *c*-axis for a *c*-plate. The spectrum at 160 °C in Figure 4 shows doublets of TA and LA modes which propagate along the *a*-axis at about 32 and 59 GHz, respectively. The broad Rayleigh wings observed at 0 GHz are a CP with A_1_(*z*) symmetry, which is related to the polarization fluctuations along the ferroelectric *c*-axis. The maximum of intensity of a CP was observed at 152 °C. The spectrum at 160 °C in Figure 5 shows TA and LA modes which propagate along the *c*-axis at about 32 and 48 GHz, respectively. However, a CP related to the polarization fluctuations in the *ab*-plane, which is perpendicular to the ferroelectric *c*-axis, was not observed at all the temperatures.

The measured Brillouin spectra were fitted using Voigt functions, a convolution of Lorentzian and Gaussian functions, for which the width of the Gaussian function was fixed as an instrumental function. The temperature dependence of frequency shift and width of the LA mode, which propagates along the ferroelectric *c*-axis, were determined by the fitting, as shown in Figure 6. Upon cooling from the high temperature above the Burns temperature, *T*_B_ = 520 °C, the LA frequency shows the remarkable softening toward *T*_c_ = 152 °C. Upon cooling from the high temperature, the LA mode width shows a remarkable increase toward *T*_c_. Such an elastic anomaly is related to the temperature evolution of polar nanoregions (PNRs) triggered by the RFs [15,23]. In the ferroelectric phase, the width gradually decreases due to the freezing of PNRs into stable nanodomains.

In lead-based relaxor ferroelectrics with perovskite structure, it is known that *T*_B_ = 427 °C and the intermediate temperature, *T** = 227 °C, are unaffected by compositions [24]. In SBN, *T*_B_ = 350 °C and *T** = 190 °C are unaffected by compositions of Sr ions, and in CBN, *T*_B_ = 517 °C and *T** = 367 °C are unaffected by compositions of Ca ions [23]. In CSBN—if we assume linear change in *T*_B_ and *T** for the composition dependence—*T*_B_ = 434 °C and *T** = 279 °C are expected for CSBN50, as shown in Figure 6. Upon cooling, the dynamic–static transition and rapid growth PNRs occur at *T**, and the remarkable decrease of the LA frequency and an increase in LA width occur towards *T*_C_ due to the scattering of LA phonons by PNRs.

The sound velocity, *V*, is determined by the frequency shift $\nu_{B}$ in the Brillouin scattering spectrum using the equation:(1)$$V=\frac{\lambda_{i}v_{B}}{2n\sin\frac{\theta }{2}}$$
where *λ_i_*, *θ*, and *n* are the wavelength of an incident beam, the scattering angle, and the refractive index of the sample, respectively. The velocity is determined from the frequency shift. The attenuation, α, is determined using
(2)$$\alpha =\frac{\pi \Gamma }{V}$$
where Γ is the FWHM of the Brillouin peak [25]. The dispersions of refractive indices and the Curie temperatures were determined in CSBN100*x* crystals grown using the Czochralski method [26]. The temperature dependences of LA velocity and LA attenuation calculated from the LA shift and width for five CSBN crystals using the values of refractive indices [26] are shown in Figure 7 and Figure 8, respectively. The LA velocity and attenuation of CBN28 with very weak RFs show remarkable changes in the vicinity of *T*_C_ = 254 °C and a sharp minimum in the LA velocity. In contrast, in SBN61, with strong RFs, the temperature dependence of the LA velocity and the attenuation in the vicinity of *T*_C_ = 72 °C are diffusive in the vicinity of *T*_C_. These differences can be caused by the variation in the strength of RFs, which suppress the sharp changes in the vicinity of *T*_C_. The temperature dependences of LA velocity and attenuation gradually change as the CBN28 content decreases from normal ferroelectric, such as CBN28, to relaxor SBN61. This crossover was also reported on the dielectric properties of CSBN ceramics [27]. Such a crossover from normal to relaxor ferroelectrics was also studied in SBN from Ba- to Sr- rich regions [7,28,29]. Recently, the crossover from normal to relaxor nature in SBN and CSBN ceramics was studied by the Rietveld refinement of X-ray diffraction and Raman spectroscopy [18]. The observed site occupancies at A1 and A2 sites were analyzed using residual entropy calculations. It was concluded that the origin of the crossover is directly related to the majority of A2 sites being occupied by Ba^2+^.

Among elastic constants, the coupling between *c*_33_ and the polarization fluctuations along the ferroelectric c-axis is very strong, while the coupling between other elastic constants and the order parameters of a ferroelectric phase transition is very week. Consequently, the elastic anomaly of other elastic constants is very small, and it is difficult to discuss the dynamical properties of a phase transition using the other two elastic constants. Therefore, the elastic anomaly of *c*_33_ was analyzed. The difference in temperature dependences of the elastic stiffness constant $c_{33}=\rho V_{LA}^{2}$ between CBN28 and CSBN75 is shown in Figure 9a,b, respectively, where ρ and *V*_LA_ are the density and LA velocity of Figure 7, respectively. In comparison with the remarkable softening of elastic constant of CBN28 towards *T*_c_, that of CSBN25 is suppressed, and the anomaly becomes diffusive. The elastic anomaly of perovskite ferroelectrics was analyzed by the following equation in the paraelectric phase.
(3)$$c_{ij}\left(t\right)=c_{ij}^{0}+c_{ij}^{1}T-c_{ij}^{2}{\left(\frac{T-T_{0}}{T_{0}}\right)}^{-n},\mathrm{for}\;T_{0}\leq T_{c}\leq T.$$

Here, $c_{ij}^{0}$, $c_{ij}^{1}$, and $c_{ij}^{2}$ are constants. On the right-hand side, the second term is the anharmonic effect and the third term is the elastic anomaly caused by the fluctuations in the order parameter. The critical exponent, *n*, was predicted to be 0.5 for three-dimensional fluctuations, 1.0 for two-dimensional fluctuations, and 1.5 for one-dimensional fluctuation of the order parameters [30]. For example, for undoped and Li-doped K(Ta_0.6_Nb_0.4_)TiO_3_ crystals with perovskite structure, the observed exponent of *n* = 0.5 indicates the three-dimensional fluctuations of polarization, which are related to the 8-site model of the off-center of B-site ions along the eight equivalent [111] directions [31]. In CBN28, the fitted value of *n* = 1.52 indicates the one-dimensional fluctuations in polarization along the [001] axis, which is suitable for a uniaxial ferroelectric phase transition. In CSBN75, the fitted value *n* = 2.17 shows the deviation from the value of theoretical model, in which structural disorder was not considered. The origin of the deviation from *n* = 1.5 in CSBN75 is probably attributed to the diffusive nature induced by the RFs.

### 3.2. Critical Slowing down Observed by a Central Peak

The relaxation process of the polarization fluctuations along a ferroelectric *c*-axis has been observed from the width of the narrow CP in CBN28 [15]. Temperature dependences of relaxation time determined from the width of a narrow central peak observed using *a*-plates of five CSBN crystals were analyzed. In CBN28, the temperature dependences of relaxation time of polarization fluctuations determined by a central peak along the *c*-axis show the critical slowing down toward *T*_C_. However, as the CBN28 content decreases, the slowing down is stretched by the strengthened RFs. In relaxor ferroelectrics, the dielectric constant ε obeys the extended Curie–Weiss law based on the compositional heterogeneity model as below [32].
(4)$$\frac{1}{\varepsilon_{}}=\frac{1}{\varepsilon_{0}}+\frac{1}{\varepsilon_{1}}{\left(\frac{T-T_{C}}{T_{C}}\right)}^{\gamma },\mathrm{for}\;T>T_{c}.$$

Here, *ε*_0_ and *ε*_1_ are constants and *γ* is the diffuseness exponent. The case of *γ* = 1 refers to the Curie–Weiss law for a normal ferroelectric phase transition, and that of *γ* = 2 is a typical relaxor phase transition. For a partially disordered ferroelectric phase transition, it holds 1 < *γ* ≤ 2. In Nb-doped Pb(Zr_0.75_Ti_0.25_)O_3_ ceramics, *γ* = 1.5 was reported [33]. The dielectric properties of CSBN crystals showed that a relaxor nature with noticeable dielectric dispersion was observed in CSBN25 [22]. The origin of its relaxor properties was attributed to lower excess oxygen in CSBN25.

In an order–disorder ferroelectric phase transition, the relaxation time *τ* shows critical slowing down towards *T*_c_. However, in the ferroelectric phase transitions of relaxor ferroelectrics, the diffusive nature was observed in the critical slowing down [34]. In 0.93Pb(Zn_1/3_Nb_2/3_)O_3_-0.07PbTiO_3_ (PZN-7PT), the slowing down was suppressed below the intermediate temperature *T** and typical critical slowing down was not observed near *T*_c_ [35]. The local transition from dynamic to static PNRs at *T** suppresses further slowing down. To describe such a suppressed slowing down by RFs, an empirical equation of the stretched slowing down was used in the vicinity of *T*_c_, as given by the following equation [35],
(5)$$\frac{1}{\tau_{CP}}=\frac{1}{\tau_{0}}+\frac{1}{\tau_{1}}{\left(\frac{T-T_{C}}{T_{C}}\right)}^{\beta },\mathrm{for}\;T>T_{c}.$$
where *β* is the stretched exponent. In the case of *β* = 1.0, Equation (5) gives the critical slowing down of normal ferroelectrics. Regarding TTB ferroelectrics, *β* = 1.0 was reported for BNN [36] and CBN28 [15]. In the case of *β* > 1.0, the slowing down of relaxation time is suppressed and/or stretched by the increase of the strength of RFs. In PMN-17PT and PMN-56PT, the observed values are *β* = 2.12 and 1.43, respectively [37]. RFs of PMN-17PT are stronger than those of PMN-56PT.

The difference in the temperature dependences of the stretched slowing down is shown in Figure 10 between CSBN75 and CSBN50 crystals. The dotted lines are values fitted using Equation (5) with *β* = 1.53 for CSBN75 and *β* = 1.88 for CSBN50 in a paraelectric phase. Therefore, as the CBN28 content decreases, the stretched exponent increases monotonically from *β* = 1 of the normal ferroelectric case.

The CBN28 content dependence of stretched index of five CSBN crystals is shown in Figure 11. The values of the diffuseness exponent determined by dielectric measurement using Equation (4) are also plotted for the comparison [27]. These exponents increase from 1.0 as the CBN28 content decreases. These similar dependences in the stretched index and diffuseness exponent indicate a crossover from a normal ferroelectric-like transition to a relaxor ferroelectric transition in CSBN.

Recently, the origin of the crossover from normal ferroelectric to relaxor in SBN and CSBN was discussed on the basis of site occupancy at A1 and A2 sites and residual entropy calculation [18]. The origin of the relaxor nature is attributed to the larger occupancy ratio of Sr:Ba at A2 sites. Differently from Sr ions, smaller Ca ions occupy only the A1 site. Therefore, the larger occupancy ratio decreases as the Ca content increases and the relaxor nature of SBN is suppressed. As another reason, lower excess oxygen in CSBN25 was also suggested [22]. To clarify the microscopic origin of the relaxor nature in TTB ferroelectrics, further studies are necessary.

## 4. Conclusions

Three dimensional relaxor ferroelectrics have been extensively studied with respect to their physical properties and to their origin of relaxor nature. Within this context, the results on uniaxial relaxor ferroelectrics have not completely been satisfactory. In this paper, the ferroelectric phase transitions of uniaxial ferroelectrics with tetragonal tungsten bronze structure were studied. Using Brillouin scattering spectroscopy, the elastic anomaly and a central peak (CP) with A_1_(*z*) symmetry were investigated. Sr*_x_*Ba_1−*x*_Nb_2_O_6_ (SBN100*x*) with strong random fields (RFs) undergoes a relaxor ferroelectric phase transition, while Ca*_x_*Ba_1−*x*_Nb_2_O_6_ (CBN100*x*) with weak RFs undergoes a nearly normal ferroelectric phase transition. The crossover from normal to relaxor ferroelectric behaviors was investigated in *x*Ca_0.28_Ba_0.72_Nb_2_O_6_-(1 − *x*)Sr_0.61_Ba_0.39_Nb_2_O_6_ (CSBN100*x*) crystals. In a CBN28, crystal, the sharp elastic anomaly of the LA mode was observed in the vicinity of *T*_C_. As the CBN28 content decreases, the anomaly of CSBN100*x* crystals becomes diffusive. A CSBN0 (SBN61) crystal shows a typical relaxor nature. In CBN28, the relaxation time determined from the width of a CP shows critical slowing down in the vicinity of *T*_C_. While, as the CBN28 content decreases, slowing down of relaxation time in the vicinity of *T*_C_ becomes diffusive. In the gigahertz range, it was observed that two dynamical processes show a crossover from normal to relaxor nature in uniaxial ferroelectric CSBN100*x* crystals. The origins of the crossover from relaxor to normal ferroelectric phase transitions are discussed with regards to the difference in the A1 and A2 sites occupancies.

## Figures and Tables

**Figure 1 materials-16-02502-f001:**
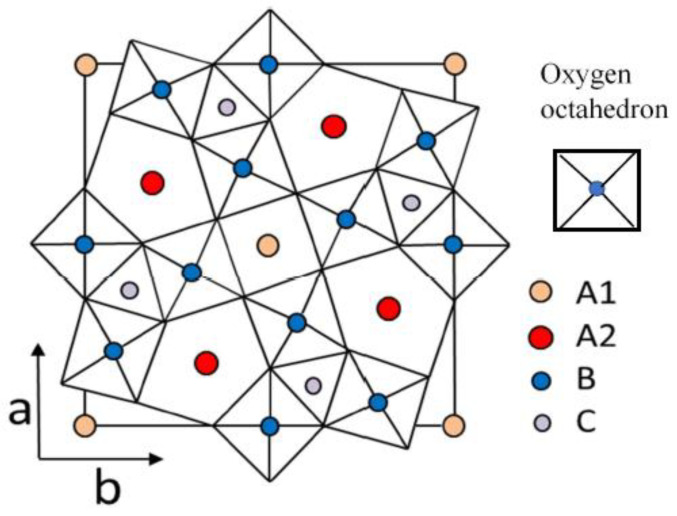
Crystal structure of *ab* plane, which is perpendicular to the polar *c*-axis, in tetragonal tungsten bronze ferroelectrics. There are three different interstices (two square A1, four pentagonal A2, and four trigonal C sites in a unit cell).

**Figure 2 materials-16-02502-f002:**
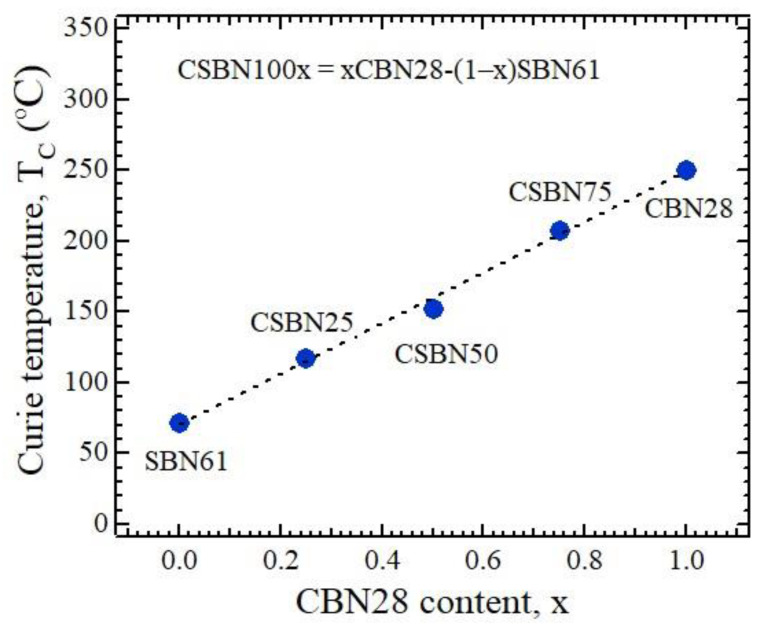
Ferroelectric Curie temperatures of calcium strontium barium niobate crystals with TTB structure. The dotted line is a linear fitting.

**Figure 3 materials-16-02502-f003:**
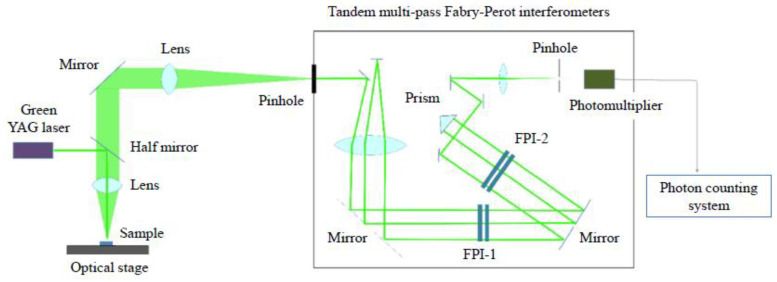
Schematic illustration of experimental setup of Brillouin scattering measurement with tandem multipass Fabry–Perot interferometers.

**Figure 4 materials-16-02502-f004:**
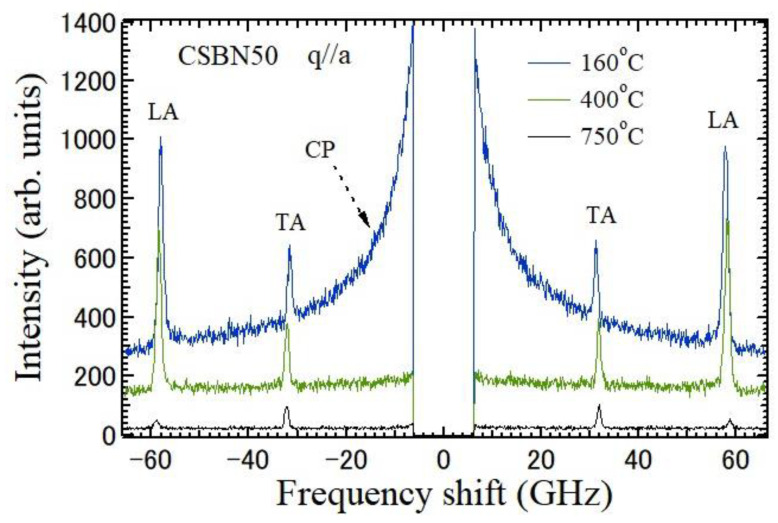
Temperature dependences of Brillouin scattering spectra measured by backward scattering geometry using the *a*-plate of a CSBN50 crystal. The wave vector *q* of the scattered phonon is parallel to the *a*-axis. Doublets of TA and LA modes which propagate along the *a*-axis were also observed. An intense CP was observed at zero frequency shift.

**Figure 5 materials-16-02502-f005:**
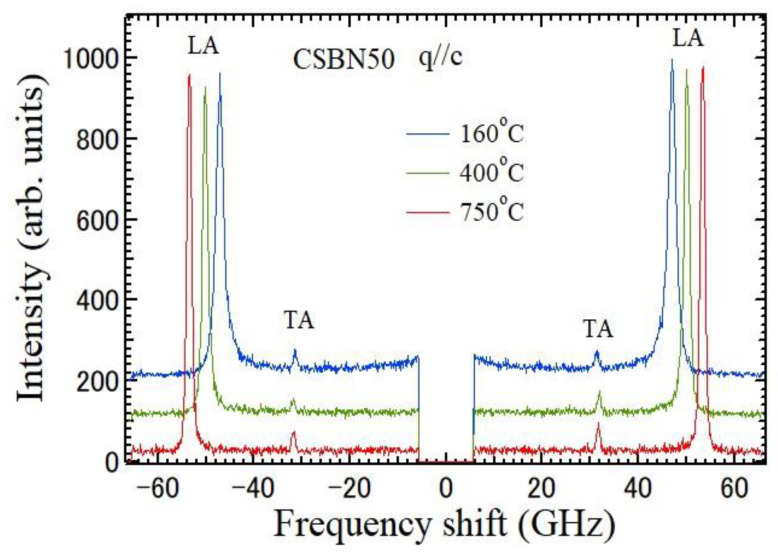
Temperature dependence of Brillouin scattering spectra measured by backward scattering geometry using the *c*-plate of a CSBN50 crystal. The wave vector *q* of the scattered phonon is parallel to the *c*-axis. Doublets of TA and LA modes which propagate along the *c*-axis were observed. No CP was observed.

**Figure 6 materials-16-02502-f006:**
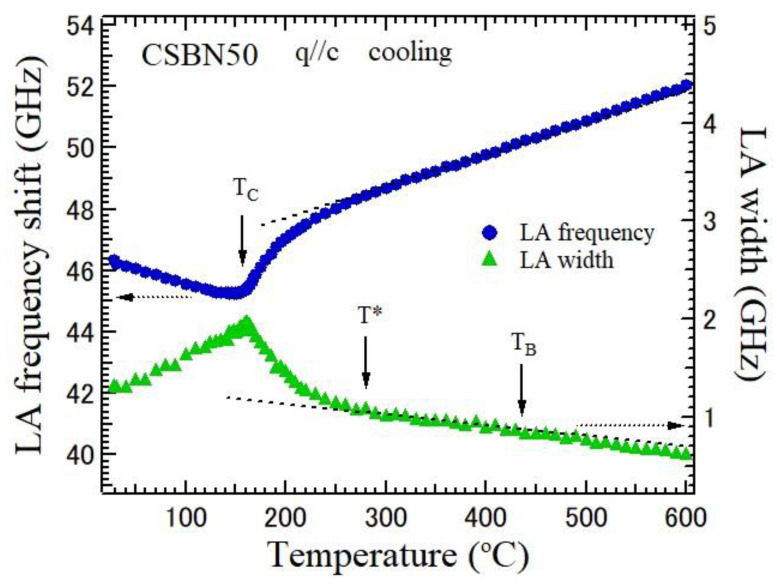
Temperature dependence of LA frequency shift and width along the ferroelectric *c*-axis of a CSBN50 crystal on cooling. The dotted lines show the nearly linear temperature dependences at high temperatures. T*and T_B_ are the intermediate and Burns temperatures, respectively. The dotted line arrows indicate the corresponding *y*-axis.

**Figure 7 materials-16-02502-f007:**
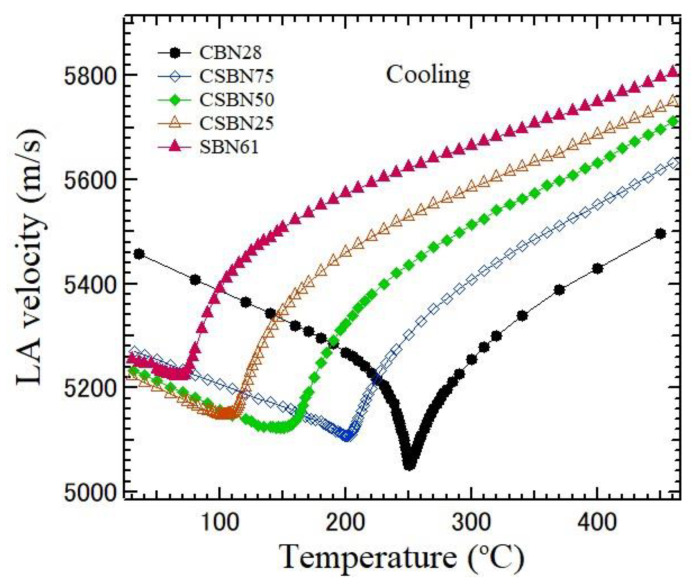
Temperature dependences of LA velocity which propagates along the ferroelectric *c*-axis of five CSBN100*x* crystals on cooling. As the CBN28 content decreases, the change in LA velocity becomes diffusive in the vicinity of *T*_c_.

**Figure 8 materials-16-02502-f008:**
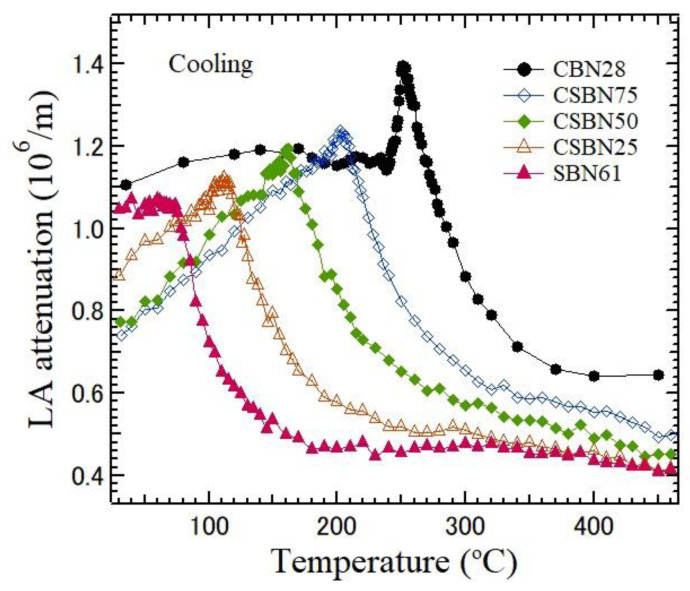
Temperature dependences of LA attenuation, which propagates along the ferroelectric *c*-axis of five CSBN100*x* crystals on cooling. As the CBN28 content decreases, the change in LA attenuation becomes diffusive in the vicinity of *T*_c_.

**Figure 9 materials-16-02502-f009:**
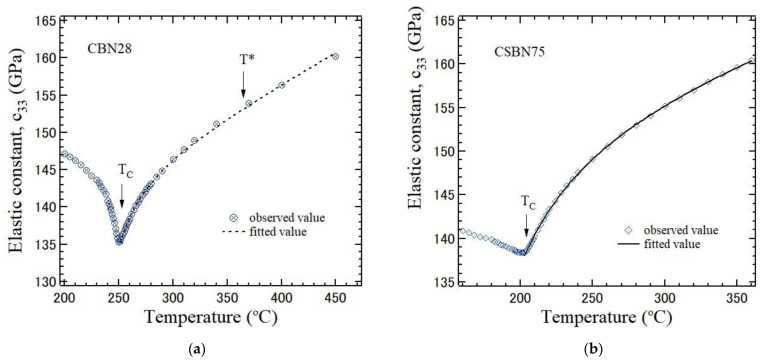
Difference in the temperature dependences of the elastic stiffness constant c_33_ between (**a**) CBN28 and (**b**) CSBN75 crystals. The elastic constant of CBN28 shows a sharp anomaly in the vicinity of *T*_C_, while that of CSBN75 shows the diffusive nature and that the elastic anomaly is smaller. The solid lines in a paraelectric phase are curves fitted using Equation (3).

**Figure 10 materials-16-02502-f010:**
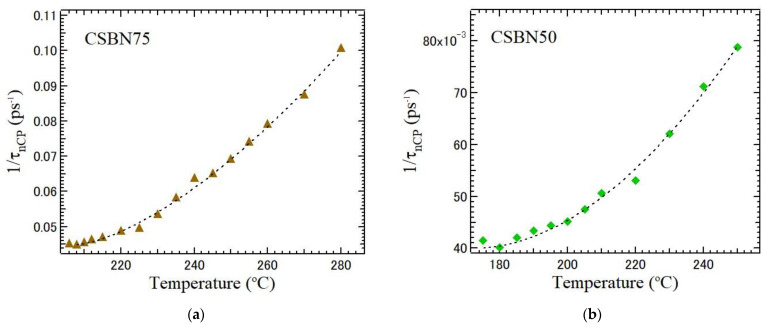
Difference in the temperature dependences of the stretched slowing down between (**a**) CSBN75 and (**b**) CSBN50 crystals. The brown triangles and green box are the observed values of CSBN75 and CSBN50, respectively. The dotted lines are fitted values to Equation (5) with *β* = 1.53 for CSBN75 and *β* = 1.88 for CSBN50.

**Figure 11 materials-16-02502-f011:**
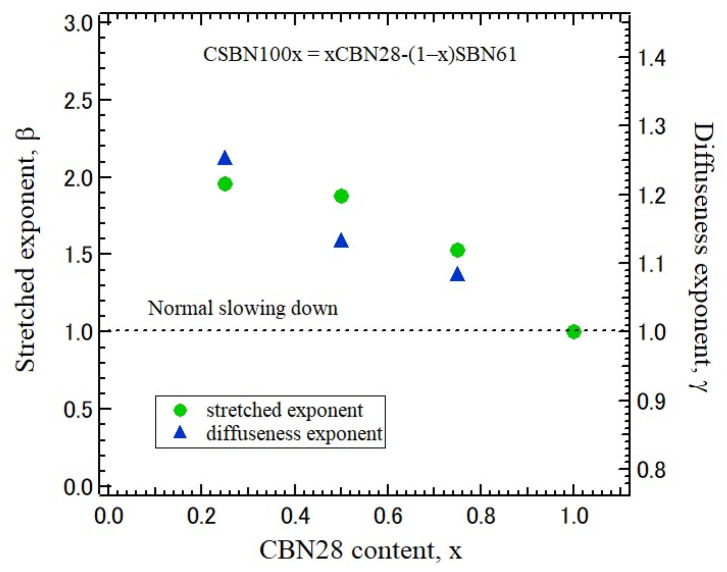
CBN28 content dependence of the exponent of stretched slowing down of Equation (5) in CSBN crystals. The values of the diffuseness exponent of dielectric constant of Equation (4) [27] are also plotted by way of the comparison.

## Data Availability

Data are contained within the articles.

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
