# Peer review of "Brillouin Scattering Study of Ferroelectric Instability of Calcium–Strontium–Barium Niobate Single Crystals"

_materials, 2023, doi:10.3390/ma16062502_

Round 1

Reviewer 1 Report

In their work, Kojima and colleagues present a systematic study of Brillouin scattering in CSBN single crystals, and the transition from relaxor to normal ferroelectric phases are discussed. 

The experimental work is sound and the analysis provided is given in enough detail, but some details are lacking, particularly in regards to structural characterization of the resulting materials used in this work.

X-ray diffraction corresponding to the samples studied should be included, at a minimum in the supplementary materials. As of now, no structural measurements are given, and it would also be useful to show the the ICP-OES data. As it stands, the statement the authors include "It has appeared that the real composition was, within uncertainties, satisfactorily close to the established one" is very vague for a scientific publication. How close is "satisfactorily close?"

The authors should note the laser power used during BLS measurements. 

Otherwise, I do believe that this work satisfies the requirements for publication in Materials and provides additional insight into the role of site occupancies on the crossover from normal to relaxor ferroelectric behavior in CSBN ferroelectrics.

Author Response

Thank you very much for the valuable comment. We have responded to every comment.

Reviewer 2 Report

I recommend this manuscript to be published when the following are addressed:

1

In the abstract, the x in 'x = 1' should not be subscript.

c in "Tc = 1" should be subscript

2

The first few sentences in first few paragraphs in Introduction do not have citations.

3

Equations (1) and (2) should be derived or cited.

4

The discussion on Figures 8 and 9 involves Tc, TB and T*, but they are not indicated on the figure

5

On page 9, in the first sentence of the second paragraph, 'sowing' should be 'slowing'

6

On page 9, how do the author define "the diffuseness index" and how it is measured.

7

How are "stretched exponent" and "diffuseness exponent" related or compared.

8

The authored mentioned a-plane and c-plane. It would be better if they can be visualized.

9

There should be a graphical representation of the experiment setup in Section 2, highlighting the special measurement techniques of ferroelectric material when compared with conventional materials.

Author Response

Thank you very much for valuable comment. We responded every comment.

Reviewer 3 Report

The authors studied the ferroelectric phase transitions of uniaxial ferroelectrics with tetragonal tungsten bronze structure by using Brillouin scattering spectroscopy. The origin of the crossover from relaxor to normal ferroelectric phase transitions is investigated. The results are sound, and the manuscript is well written. I can recommend its publication after the following issues are addressed: 

1.      The elastic constant includes three components, while there only give one of them. How about the other two?

2.      About figure 7 of the LA attenuation, why the addition of SB shows unique compared with SB61 and CBN28

3.      There are also some English grammar issues.

Author Response

Thank you for your valuable comment. We responded to every comment.
